# Bullying in Adolescents: Differences between Gender and School Year and Relationship with Academic Performance

**DOI:** 10.3390/ijerph19159301

**Published:** 2022-07-29

**Authors:** Ana Isabel Obregón-Cuesta, Luis Alberto Mínguez-Mínguez, Benito León-del-Barco, Santiago Mendo-Lázaro, Jessica Fernández-Solana, Jerónimo J. González-Bernal, Josefa González-Santos

**Affiliations:** 1Department of Mathematics and Computing, University of Burgos, 09001 Burgos, Spain; aiobregon@ubu.es; 2Department of Education Sciences, University of Burgos, 09001 Burgos, Spain; 3Department of Psychology and Anthropology, University of Extremadura, 10071 Cáceres, Spain; bleon@unex.es (B.L.-d.-B.); smendo@unex.es (S.M.-L.); 4Department of Health Sciences, University of Burgos, 09001 Burgos, Spain; jfsolana@ubu.es (J.F.-S.); jejavier@ubu.es (J.J.G.-B.); mjgonzalez@ubu.es (J.G.-S.)

**Keywords:** bullying, students, primary education, secondary education, gender, academic year, academic performance, categories

## Abstract

School bullying is a phenomenon of unjustified aggression in the school environment that is widespread throughout the world and with serious consequences for both the bully and the bullied. The objectives of this research were to analyze the differences between the different bullying categories by gender and academic year in primary and secondary education students, as well as their relationship with academic performance. To categorize students according to their bullying experiences, the European Bullying Intervention Project (EBIPQ) Questionnaire was used. The Chi-square test was used to compare the scores obtained by the students in the EBIPQ based on gender and academic year, and the one-way ANOVA test was used to analyze its relationship with academic performance. Research participants were 562 students from the 5th (*n* = 228) and 6th (*n* = 186) primary school years and the 1st (*n* = 134) and 2nd (*n* = 94) secondary school years. They were males (50.5%) and females (49.5%) ranging in ages from 10 to 15 years old (mean = 11.66, standard deviation = 1.206). The results showed statistically significant differences in gender and academic year, indicating a greater number of boys in the role of the bully/victim and girls in that of non-bully/non-victim. The most aggressive students were in the 2nd year of ESO (12–13 years old). Regarding academic performance, statistically significant differences were obtained that confirm the hypothesis that performance or average grade varies according to the category of bullying in which students find themselves. The academic performance of the non-bully/non-victim and those in the victim category was found to be higher than that of bullies and bully/victim students.

## 1. Introduction

Although there is no standard definition of bullying, a lot of research has been conducted in this area. An early definition referred to bullying as “a student being victimized or bullied when exposed repeatedly and over time to negative actions by one or more students”; later, more precise definitions referred to the manner in which the student could be assaulted [1]. In more recent definitions, bullying has been defined as deliberate aggression or intentional harm-doing carried out by one or more people repeatedly and over time in an interpersonal setting, characterized by an imbalance of power, either real or perceived, that makes it difficult for the victims to defend themselves from the aggressors. [2,3,4]. Studies have found that people who differ from “the majority” in some way, such as sexual orientation, gender, race, presence of pathologies, etc., are particularly vulnerable to victimization [1].

It is possible to distinguish bullying from sporadic fights among young people who are in similar conditions of physical, psychological or social strength by using this definition [5,6]. To some extent, definitions of bullying reflect how they are used, i.e., in the case of the definition of bullying itself, it not only describes the consensus on what it is or is not but also informs how to recognize, prevent and stop it [7]. The term “bullying” and its terminology are also debated in research [7,8]; however, by definition, it is a serious and pervasive problem that severely affects those exposed to it [9]. A lack of clear and standardized definitions of bullying impair progress in understanding this serious and complex issue [1]. It should also be noted that today, bullying behaviors have spread to social networks, the internet and mobile phones; therefore, disagreements have been observed in the literature regarding the definition of bullying, linked to discipline and culture [8]. It should be noted that cyberbullying is described as harmful acts that are intentionally performed and inflicted through electronic devices. However, this term cannot be considered a unidimensional construct as it includes a wide range of online experiences such as exclusion, cyberbullying, denigration, or impersonation [10].

At present, bullying is widespread worldwide, with a high prevalence in all countries, constituting a public health problem [3,11], which has led to an increase in the number of studies on the subject in recent decades [4]. In many countries, rates of bullying are very high; according to a study of adolescents aged 11–15, the prevalence of bullying victimization (a term indicating that a person is a victim of bullying) has indicated an average rate of over 17% in 25 countries in Europe and North America [12]. Likewise, it is considered one of the main health problems in childhood and adolescence according to the World Health Organization (WHO) [7], since almost one-third of primary and secondary school pupils are involved in bullying, in addition to the role of victims and aggressors, various other roles such as bystanders or people who reinforce/defend the victim/bully [13].

Research on bullying has focused, until recent years, on the bully/victim binomial, with much less attention paid to the bystander role. The reason why the bystander role has been less explored is its complexity, due to the numerous and conflicting strategic factors that lead to its adoption in a bullying context [14]. According to Salmivalli et al. [15], there are several roles that can be defined. These can be divided into: the role of the bully, the victim, the bully/victim, who can act as a helper of the harasser, reinforcing them, or a defender of the victim, and finally the spectator role or audience [14,16,17,18]. The idea of a static role is discarded as the same person can assume different roles throughout the year [5,11]. Bullying can be considered as a group phenomenon in which most children in the school class have a defined role [9]. Within these categories, around 35% of students are in the role of victim, with another 35% in the role of bullies. At the same time, higher rates of bullying have been observed among boys compared to girls, and higher rates also in primary school compared to the upper grades [5,19,20].

School bullying, considered a common social phenomenon worldwide, can have serious consequences, ranging from physical symptoms to significant psychiatric symptoms such as depression, anxiety, insomnia, self-harm, etc., also including poor academic performance. Likewise, it is also associated with problems in social relationships and impaired quality of life [4,21,22,23,24,25]. Physical, verbal and social abuse is a major health problem for students; those who are victims may have poorer emotional, social, academic and health development, while bullies tend to display delinquent and aggressive behaviors in late adolescence [26,27]. The type of violence in bullying varies by gender, showing a higher risk of physical victimization among boys and emotional, psychological, or relational victimization among girls [22].

Involvement in bullying, both from the bully’s and the victim’s point of view, has also been shown to be associated with negative outcomes among students, with problems of internalizing, externalizing and even dropping out of school. Moreover, the greater their involvement, the greater negative consequences are appreciated with respect to academic performance [21,22,28]. Previous research suggests that students who experience peer victimization are at increased risk of poor academic adjustment [29]. Likewise, the Nakamoto and Shwartz meta-analysis found a significant association between victimization and academic underachievement [30]. The association between bullying and mental health is also very well established, with differentiated patterns between bullies, victims, and bully/victims. Victims have been shown to have more internalized problems (such as anxiety, depression) than their peers. In contrast, bullies are more often characterized by externalizing problems (such as conduct problems and antisocial behavior), and bully-victims assume both problems, internalizing and externalizing [31,32]. Regarding the presence of bullying in school, a high presence has been observed during the school years, and also a rate that increases in the early childhood stage and decreases until the end of adolescence [31].

In terms of bullying at school, it is important to mention, ultimately, cyberbullying through electronic communication devices. Electronic bullying offers a very different problem to “traditional” bullying, as the aggression through electronic devices helps to protect the anonymity of the aggressors and, in many cases, the aggressor is not aware of the consequences of his actions on the victim, making it very difficult to empathize with the victim. These aggressions can take place anywhere and at any time, which complicates adult supervision and control [33]. This cyber aggression has also been negatively associated, not only in victims but also in aggressors, with an increase in internalized problems such as depressive symptoms in adolescents, with serious repercussions on concentration and academic performance [34]. However, it is very difficult to determine the prevalence of cyberbullying, although published studies suggest a progressive trend [33].

The objectives of this study were to establish the differences between gender and academic year with the presence of bullying and its different categories, as well as to determine if there is a relationship between suffering or exercising bullying with the level of academic performance in a sample of primary and secondary school students.

## 2. Materials and Methods

### 2.1. Participants

The sample consisted of 562 students of Compulsory Primary Education (EPO) and Compulsory Secondary Education (ESO), aged between 10 and 15 years (Mean = 11.66; Standard Deviation = 1.21), distributed in 284 (50.5%) subjects that were boys and 278 (49.5%) girls. EPO students (*n* = 334) were in fifth (*n* = 228) and sixth (*n* = 186) grades, and ESO students (*n* = 148) were in the first (*n* = 134) and second year (*n* = 94). The sample was collected from students of 5 different schools, public (*n* = 4) and private (*n* = 1), in the Autonomous Community of Castilla y León.

In this region, there is a School Co-existence Observatory, a consultative and supportive body for the entire educational community of educational centers.

There is a protocol of action against school bullying and a specific anti-bullying program, named SSR (Stop the bullying, Support the victim and Reeducate the aggressor, P.A.R. in Spanish), which includes actions to enhance relationships in the educational community.

Despite this, over 65% of Castilla y León schools have cases of bullying.

The sample selection was made by conglomerates.

### 2.2. Instruments

The European Bullying Intervention Project Questionnaire (EBIPQ) is a validated instrument translated into Spanish [35] that allows us to identify the prevalence of involvement in bullying of the aggressor, the victim and the victimized aggressor; as well as the typology (direct or indirect) and the manifestations (physical, verbal and relational) [36]. This instrument has very good psychometric properties in several European countries, including Spain [8,10].

It consists of 14 items, 7 describing aspects related to victimization and 7 in correspondence with aggression. For both dimensions, the items refer to actions such as hitting, insulting, threatening, stealing, saying foul words, and excluding or spreading rumors. The scale was designed to assess the frequency of aggression or victimization, so the items are related to the different types of bullying [7,37].

The frequency is taken into account, taking as a reference the previous two months and is evaluated by a Likert scale of 1 to 5, where the possible answers are: No; Yes, once or twice; Yes, once or twice a month; Yes, about once a week and Yes, more than once a week [36,38].

### 2.3. Procedure

Following the American Psychological Association’s ethical guidelines of consent, confidentiality, and anonymity in the responses, school principals were first contacted, and the aims of the research were explained to them.

Each principal, with the support of researchers, supervised the communication of the purpose of the study to the parents and the collection of informed consent.

Once the collaboration was accepted, the participants were contacted in the classroom, and after receiving informed parental consent, they were given the opportunity to fill in the scales freely. It was carried out anonymously, ensuring the confidentiality of the data obtained and its exclusive use for research purposes. The administration of the scales was carried out during school hours, providing relevant instructions. The anonymous nature of the research was highlighted. The questionnaires were filled out individually in a suitable environment and without distractions. The process of filling the questionnaires took about 15 min.

No data was lost.

No questionnaire was rejected.

The principals selected the grades that would be included in the study, and all the students in those classes were invited to participate in the research. No boy or girl was excluded based on their culture, language, religion, race, disability, sexual orientation, ethnicity, gender, or age.

The Bioethics Committee of the University of Burgos approved the research, (Reference UBU 032/2021), respecting all the requirements established in the Helsinki Declaration of 1975.

### 2.4. Data Analysis

To compare the scores obtained in the EBIPQ according to gender and academic year, the Chi-square test was used, while the analysis according to school performance was carried out using the one-factor ANOVA test. To study the correlation between the different categories of bullying obtained with the EBIPQ scale and school performance, Pearson’s test was used. A statistical significance value of *p* < 0.05 was established using SPSS version 25 software (IBM-Inc., Chicago, IL, USA).

## 3. Results

### 3.1. Associations between Bullying Categories, Gender and Academic Year

The results of the inferential analysis showed statistically significant differences in the categorization of subjects into types of bullying according to gender (χ^2^ = 12,960; *p* = 0.005).

There were more girls (62.6%) classified as non-victim/non- bully than boys (52.1%), and there were more boys in the bully/victim role (18.7%) than girls (10.1%) (Table 1).

The Chi-square test showed a statistically significant relationship between bullying categories and the academic year of the students (χ^2^ = 16,896; *p* = 0.05).

Significant differences between observed and expected frequencies were found for the 2nd ESO pupils in the category of bully, with more being classed as bullies (8.5%) than in the rest of the academic years (2–3%), and the 5th grade of primary school pupils in the category of bully/victim, with more being classed as bullies/victims (20.9%) than in the rest of the academic years (10.8–13.4%) (Table 2).

### 3.2. Comparation between Bullying Categories and Academic Performance

A statistical significance was observed (*p* > 0.001) between bullying categories and academic performance.

Academic performance was obtained by calculating an average of the grades obtained by each student in all the subjects corresponding to the last assessment carried out.

The hypothesis that the performance or the average mark varies according to the categorization of students in terms of bullying was confirmed: there are significant differences in students’ performance according to this categorization (F_(3,558)_ = 7.319, *p* < 0.001).

The one-factor ANOVA test (Table 3) showed that the lowest performance is obtained by bullies/victims 6.82 (SD = 1.38), followed by bullies 7.01 (SD = 1.61), victims 7.331 (SD = 1.38), and finally the highest performers are non-victims and non-bullies 7.56 (SD = 1.27).

A post hoc analysis was performed to compare the averages in pairs of each of the categories of bullying.

It was observed that the comparison between students in the non-victim/non-bully category and the students in the bully/victim category (*p* < 0.001) and a comparison between students in the victim category and students in the bully/victim category (*p* = 0.007) was significant. It can be concluded that students in the non-victim/non-bullying category and students in the victim category have a higher academic performance than students in the bully/victim category.

## 4. Discussion

School bullying is widespread globally and shows a large-scale prevalence in studies in all countries, ranging from a low of 10% to a high of 70% [4], despite the fact that anti-bullying protocols are in place in almost all regions.

Our study showed that forty-three percent of children surveyed were somehow involved in bullying either as a victim, bully, or both.

Considering a global perspective, the main contribution of the study is to provide information on how bullying is distributed across different ages and genders and how it affects academic performance. This information will be used to develop more specific anti-bullying interventions in the future.

The first aim of the study was to establish the relationship between the presence of bullying and students’ gender and academic year.

The results of our study show the existence of a significant relationship between the categorization of students according to the types of bullying and their gender. Specifically, this relationship was observed for the categories of bully/victim and no-victim-no-bully in boys, finding a greater number than expected for this gender in the role of bully/victim and a smaller number for the role of the non-victim/non-bully. At the same time, the opposite results were obtained for the female gender, where we also found significant differences in the roles of bully/victim and non-victim/non-bully. There are no significant differences for both genders in the roles of victim or bully.

According the research by Ordoñez-Ordoñez et al. [39], boys and male adolescents are the ones with the greatest involvement in bully and bully/victim roles, corroborating the results of our study where a greater number of boys were found in the bully/victim role, 18.7%, compared to 10.1% in girls. Other studies have found a higher prevalence of bullies and bullies/victims among males and, at the same time, of victims among females [14,36,38,40,41]. In contrast, the results of the study by Górriz et al. [17] show significant findings regarding a greater presence of the male gender in the victim role and the female gender in the bully role.

Types of behaviors related to bullying are also differentiated according to gender, with physical violence, insults or threats being the most experienced among boys; while girls are related to relational behaviors such as exclusion, spreading rumors or being ignored by other peers [19,42,43]. The differences in the figures and behaviors found for both genders can be explained by taking into account gender socialization and normative expectation of both, understanding bullying as a behavior in which different genders act according to what is expected of them [19,43].

Bullying allows the aggressor to demonstrate his or her physical strength, dominance over others, and rank in the social hierarchy [44]. In this way, and according to the existing evidence, bullying has a dual direction, in which a person who has been a victim may later become a bully, making both roles risk factors for future aggressions [45]. Jing Wang et al. [46] confirm gender differences in direct and indirect forms of bullying.

These findings suggest that boys and girls take on social stereotypes. Thus, the masculine stereotype associated with virility and violence as opposed to femineity is therefore assimilated from childhood.

Consequently, it seems reasonable that anti-bullying interventions should be aimed at breaking down sexist social stereotypes.

Regarding the academic year and bullying, our study also found a significant relationship, with 2nd ESO students assuming the role of bully in a higher proportion than the rest of the years, and 5th grade primary school students assuming the highest percentage of bully/victim. Our results are partially confirmed in the Benítez-Sillero et al. [47] study, where higher results were found for the bully role between 11 and 15 years of age; that is, between 1st and 4th of ESO, as well as the study by Herrera-López et al. [36] where it is highlighted that the students with the greatest involvement are found in the middle years of secondary school, and then decrease in the upper years. These findings contrast with those of other research, which found no significant differences in the different roles concerning the student’s age [17,39].

It is important that anti-bullying interventions are carried out from an early age to prevent bullying in future ages. However, given that our study and others show that there is an age group in which it is more widespread, it would be interesting to carry out specific actions aimed at pupils in those school years (1st–4th of ESO).

In the second objective of our study, the existence of a relationship between the academic performance of students with respect to suffering or practicing bullying was considered. Understanding academic performance as the average mark obtained in the last exams in all subjects, the results showed significant differences between the mean grades of the different categories, thus confirming the hypothesis that bullying is related to the variation in the grades of the academic record [48]. It is worth noting that the lowest level ratings are associated with bully/victim behaviors, followed by the bully role, which can be explained or justified by the characteristics of the group to which they tend to belong. In other words, finding yourself in these roles within bullying means higher levels of Machiavellianism and being perceived as popular among peers, although Mariko Hosozawa [49] finds that students with lower academic performance were more likely to be victimized.

Our study shows that the students in the non-victim/nno-bully role, followed by the victim role, presented the highest academic achievement scores; however, it has been shown that the latter often manifest school avoidance behaviors. This can manifest both short- and long-term consequences, causing the loss of important learning and social opportunities, thus significantly hindering academic success and encouraging school dropout [11,50].

According to the research of Aparisi et al. [51], bullies, as well as bullies/victims, have greater difficulties in organizing and planning studies, in performing adequately in exams and develop fewer control and consolidation strategies than students who are neither bullies nor victims. In contrast, Clemmensen et al. [52] show in their results that the difference between bullies, bullies/victims and the uninvolved were not significant but they were significant in the case of victims, who revealed lower scores than those who were classed as non-victim/non-bully.

The learning process, review and comprehension strategies can also be affected, as well as problems in concentration, which can be associated with a rejection of school and undervaluing studies [42,52].

These results, in agreement with those found by Khanh-Dao Le et al. [44] in their research, indicate an urgent need to establish interventions to prevent both victimization and aggression in bullying in school-age children.

School bullying, both at the student level (i.e., bullying victimization) and at the school level (i.e., bullying climate), has significant and negative effects on academic performance. Bullying on campus not only affects the academic performance of bullied students but can also impair the learning of all students by shaping a negative school climate [53]. This way, all interventions aimed at preventing bullying will be positive for the academic performance of all students, not just victims and bullies [54].

Evidence shows that the most effective anti-bullying interventions are those that emphasize the prevention of violence and, more importantly, promote positive coexistence and a school culture of proper treatment and respect. However, educational institutions generally seek to develop strategies to combat bullying when it is already present in the institution; therefore, they are reactive rather than proactive measures [55].

Finally, regarding the limitations of the study, it should be pointed out that the results may not be generalizable to all children and adolescents in Spain or other countries, since the sample has been taken from a single autonomous community. Likewise, it was carried out only with children and adolescents of very specific ages, so the age range could be extended. The use of self-report questionnaires such as the EBIPQ can also be a limitation in the research, since these questionnaires must be interpreted with caution, although it is a questionnaire with good psychometric properties and validated in Spain. An important point of the study is the need to continue researching interventions that can help alleviate this situation.

## 5. Conclusions

In the present study, we found statistically significant differences with respect to bullying categories in terms of students’ academic performance (where the lowest performers are bullies/victims and the highest performers are non-victims and non-bullies), as well as in terms of gender (with more girls classed as non-victim/non-bully than boys and with more boys in the bully/victim role than girls) and academic year (students in the 2nd year of ESO were more frequently in the bully role and students in the 5th grade in the bully/victim role). This is current information that may be relevant for the implementation of interventions aimed at preventing bullying from continuing to be perpetuated to a point where it may have an impact both at the school level, worsening the levels of academic performance or school dropout, and at the level of the mental health and quality of life of children and adolescents. Future research on bullying should be complemented by new forms of harassment such as cyberbullying, which is becoming increasingly prevalent.

## Figures and Tables

**Table 1 ijerph-19-09301-t001:** Comparison of bullying categories based on gender.

			Bullying Category				
			No Victim No Bully	Victim	Bully	Bully/Victim	Total
Gender	Boy	Count	148	69	14	53	284
		Expected Count	162.7	70.2	10.1	40.9	284.0
		% within Gender	52.1%	24.3%	4.9%	18.7%	100.0%
		Adjusted Residual	−2.5	−0.2	1.8	2.9	
	Girl	Count	174	70	6	28	278
		Expected Count	159.3	68.8	9.9	40.1	278.0
		% within Gender	62.6%	25.2%	2.2%	10.1%	100.0%
		Adjusted Residual	2.5	0.2	−1.8	−2.9	
Total		Count	322	139	20	81	562
		% of Total	57.3%	24.7%	3.6%	14.4%	100.0%

**Table 2 ijerph-19-09301-t002:** Relationship between bullying categories and school year.

			Bullying Category				Total
			No Victim No Bully	Victim	Bully	Bully/Victim	
Curso	1º ESO	Count	83	29	4	18	134
		Expected Count	76.8	33.1	4.8	19.3	134.0
		% within academic year	61.9%	21.6%	3.0%	13.4%	100.0%
		Adjusted Residual	1.2	−1.0	−0.4	−0.4	
	2º ESO	Count	52	22	8	12	94
		Expected Count	53.9	23.2	3.3	13.5	94.0
		% within academic year	55.3%	23.4%	8.5%	12.8%	100.0%
		Adjusted Residual	−0.4	−0.3	2.8	−0.5	
	5º Primary	Count	78	36	3	31	148
		Expected Count	84.8	36.6	5.3	21.3	148.0
		% within academic year	52.7%	24.3%	2.0%	20.9%	100.0%
		Adjusted Residual	−1.3	−0.1	−1.2	2.6	
	6º Primary	Count	109	52	5	20	186
		Expected Count	106.6	46.0	6.6	26.8	186.0
		% within academic year	58.6%	28.0%	2.7%	10.8%	100.0%
		Adjusted Residual	0.4	1.2	−0.8	−1.7	
Total		Count	322	139	20	81	562
		% of Total	57,3%	24.7%	3.6%	14.4%	100.0%

**Table 3 ijerph-19-09301-t003:** Descriptive statistics of the comparison of bullying categories with academic performance.

Average Score								
	N	Mean	Std. Deviation	Std. Error	95% Confidence Interval for Mean		Minimum	Maximum
					Lower Bound	Upper Bound		
No victim no bully	322	7.566	1.2701	0.0708	7.427	7.705	2.5	9.8
Victim	139	7.331	1.3827	0.1173	7.099	7.563	3.1	10
Bully	20	7.011	1.6182	0.3618	6.254	7.769	2.8	9.5
Bully/victim	81	6.829	1.3870	0.1541	6.522	7.136	3.2	9.5
Total	562	7.382	1.3511	0.0570	7.270	7.494	2.5	10

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
