# Peer review of "Bullying in Adolescents: Differences between Gender and School Year and Relationship with Academic Performance"

_ijerph, 2022, doi:10.3390/ijerph19159301_

Round 1
Reviewer 1 Report
Thank you for submitting this paper which contains some interesting points about bullying and young people. Overall, the paper would benefit from thorough proofreading.
Introduction
I think it would be helpful to refer to the literature where the academic definition of bullying is critiqued. For example, Eriksen (2018), Espelage, D. L. (2018) and O’Brien (2019) (see references below). These papers and others suggest that researching bullying can be problematic mainly linked to difficulties in the definition.
Eriksen, I. M. 2018. The power of the word: students’ and school staff’s use of the established bullying definition. Educ. Res. 60, 157–170. doi: 10.1080/00131881.2018.1454263
Espelage, D. L. (2018). Understanding the complexity of school bully involvement. Chautauqua J. 2:20.
O’Brien, N., 2019. Understanding alternative bullying perspectives through research engagement with young people. Frontiers in Psychology, 10, p.1984.
Lines 91-95: It is unclear as to the relevance of this in the context of the study reported here.
In the introduction, concepts and arguments are not developed strongly enough. It reads like bullet points in some parts and expanding this would be helpful.
Materials and Methods
Participants
More details are needed about the participants and the process. How was informed consent received from the students? Were parents aware of the research? Did students have an alternative option if they did not want to participate in the survey? What was the return rate on the survey?
Line 104/105: It is unclear what is meant by: “Distributed in 284 subjects belonged to the male gender (50.5%) and 278 to the female (49.5%).” And line 107: “The sample was collected from 5 different centers,…” Please explain what these centres are.
Procedure
Line 138: “No data was lost as the data was collected using an online form in which all responses are mandatory.” There are issues around making all of the questions mandatory including issues of coercion if students did not want to answer a particular question. Could you please explain why you applied this mandatory approach?
Line 141: “All the students of the grades selected by the principals were included in the research.” This is not clear. Does this relate to the academic gradings of the students? It states under ‘participants’ that students came from two year groups in primary school and two year groups in secondary school. Was this the decision of the principals (of the centres?) or the researchers?
It is not clear where the data came for ‘academic performance’ so it is difficult to appraise these results.
Discussion
Overall, the discussion needs to be reconsidered. What is it that your study is contributing to our knowledge on this topic? For example, it is not clear why parental practices are being referred to as this was not measured on the questionnaire. If it explains your results and the contribution then it needs to be further discussed as well as the other results from the study and what they mean.
Lines 234-243 would benefit from more contextual information earlier on. What type of schools did these students attend? Is bullying a problem for the schools? Do they have anti-bullying initiatives? Are the students over/under performing academically?
Lines 268-270: “The need to continue researching interventions that can help alleviate this situation is raised as an important point of the investigation.” However, this study did not explore interventions? Perhaps adding this into the literature review earlier on and picking it up in the discussion could help with this argument.
Author Response
Response to Reviewer 1: First, we would like to express our sincere gratitude for all comments and suggestions received from the Reviewer 1. This information has certainly enriched the text for its best understanding, thank you very much indeed. We have introduced the required changes both in our answers to the specific comments and in the final manuscript. INTRODUCTION: Refer to the literature where the academic definition of bullying is critiqued. Response: Thank you very much for pointing it out. We have introduced the problems associated with the academic definition of bullying, considering proposed references. This new information is here: “There has been a great deal of research on bullying, although there is no standard definition. An early definition referred to "a student being victimised or bullied when exposed repeatedly and over time to negative actions by one or more students"; later more explicit definitions referred to the way in which the student may be assaulted [1] . In more recent definitions, bullying has been defined as deliberate aggression or intentional harm-doing carried out by one or several people repeatedly and over time in an interpersonal setting characterized by an imbalance of power, either real or perceived, that makes it difficult for the victims to protect themselves from the aggressors. [2–4]” “It is important to note, though, that definitions of bullying, to some extent, are a reflection of the way in which the terms are used, i.e. in the case of the definition of bullying itself, it not only describes the consensus on what it is or is not, but informs how to recognise, prevent and stop it [7]. Likewise, the term "bullying" and its terminology is under debate in research [7,8]; however it is defined, it is a serious and pervasive problem that severely affects those exposed to it [9]. The lack of a clear and standardised definition of the term may be an obstacle to progress in understanding the serious and complex problem of bullying [1]. It should also be noted that nowadays, bullying behaviors has spread to social networks, internet use and mobile phones; therefore, disagreements have been observed in the literature regarding the definition of bullying, linked to discipline and culture [8].” Psychological well-being and subjective happiness Response: Thank you very much for this comment. We have removed the paragraph. Concepts and arguments are not developed strongly enough Response: Thank you very much for the suggestion. We have reviewed the introduction and developed in greater depth the most relevant aspects, taking into account your comments. MATERIAL AND METHODS: Participants Response: Thank you very much for pointing it out, we've added more information about the participants. We expect that it is clearer now. “The sample consisted of 562 students of Compulsory Primary Education (EPO) and Compulsory Secondary Education (ESO), aged between 10 and 15 years (Mean=11.66; Standard Deviation=1.21). Distributed in 284 (50.5%) subjects that were boys and 278 (49.5%) girls. EPO students (n=334) were in fifth (n=228) and sixth (n=186) grades, and ESO students (n=148) were in the first (n=134) and second year (n=94). The sample was collected from students of 5 different schools, public (n = 4) and private (n = 1), in the Autonomous Community of Castilla y León.” Participants and procedure: more detail Response: Thank you very much for this comment. The information about the participants and the process has been detailed more precisely in Procedure section. Here it is the information: “ Following the ethical guidance of the American Psychological Association regarding consent, confidentiality and anonymity in the responses, the directors of the educational centers were first contacted and explained the objectives of the research. Each principal, with the support of researchers, supervised communicating the purpose of the study to the parents and collecting the informed consent. Once the collaboration was accepted, the participants were contacted in the classroom, and after receiving informed parental consent, they were given the opportunity to fill in the scales freely. It was done anonymously, ensuring the confidentiality of the data obtained and their exclusive use for research purposes. The administration of the scales was carried out during school hours, providing relevant instructions. The anonymous character of the investigation was highlighted. The questionnaires were filled out individually in a suitable environment and without distractions. The process of completing the questionnaires took about 15 minutes. No data was lost. No questionnaire was rejected. The principals selected the grades that would be included in the study, and all the students in those classes were invited to participate in the research. No boy or girl was excluded based on their culture, language, religion, race, disability, sexual orientation, ethnicity, gender, or age. The Bioethics Committee of the University of Burgos approved the research, (Reference UBU 032/2021), respecting all the requirements established in the Declaration of Helsinki of 1975” PROCEDURE: Mandatory responses Response: Thank you very much for this comment. As we inform in the Bioethics Committee of the University of Burgos report (Reference UBU 032/2021), only the students that have their tutor’s authorization could complete the questionaries. These students were offered the possibility to complete the European Bullying Intervention Project Questionnaire (EBIPQ) freely. There was no coercion to students. The sentence was only to introduce that there were not data lost because all the students that freely completed the questionaries completed the 14 questions of the scale. All the students of the grades selected by the principals were included in the research Response: Thank you very much for pointing it out. We have tried to clarify this aspect in Procedure subsection. The researches decided to carry out the study with students belonging to the 4 grades mentioned, on the one hand to study if there are differences between different grades in the same educational stage (primary or secondary school), and on the other hand to study if there are differences between the students of one stage and those of the other. After exposing this aspect to the schools’ principals, they chose the grades (of the 4 possible) would participate in the study. Academic performance Response: Thank you very much for this comment. We have included an explanation in order to be more specific: “The academic performance was obtained by making an average of the grades obtained by each student in all the subjects corresponding to last evaluation carried out.” Specifically, they belonged to the second evaluation of the school year 2020-2021. DISCUSSION Discussion reconsidered Thank you very much for this suggestion. We revised the whole Discussion section by highlighting and detailing the major contributions of the study and eliminating the less relevant aspects. We hope it is clearer now. Contextual information about participants Thank you very much for pointing it out. We have oncluded all the points you asked for in Participants section. The new information is: “The sample was collected from students of 5 different schools, public (n = 4) and private (n = 1), in the Autonomous Community of Castilla y León. In this region, there is an Observatory for School Co-existence, a consultative and supportive body for the entire educational community of educational centers. There is a protocol of action against school bullying and a specific anti-bullying program, P.A.R. which includes actions to enhance relationships in the educational community. Despite this, over 65% of Castilla y León schools have cases of bullying.” Interventions Response: Thank you very much for pointing it out. We have included more information about interventions already done and about how future interventions should be.
Reviewer 2 Report
I was pleased to review this work which situates within the bullying research field. The study analyzed differences between the different bullying categories across gender and the academic year among 562 students of different grades. The results showed a greater number of boys in the role of bully/victim and girls in that of non-bully-nor-victim; also the most aggressive students were in 2nd year of ESO (12-13 years old). Finally students who were non-bully-nor-victim and those in the victim category had a higher academic performance than the bulllies and bully/victim. Overall, the study was well conducted and presented. I have only a couple of minor issue to be addressed to strengthen the manuscript:
- In the abstract it seems that something is missing here: "Age between 10 and 15 years old (mean=11.66, standard deviation=1.206) both males (50.5%) and females (49.5%)"
- There is no mention of cyberbullying in the Introduction, but it is only mentioned as a future direction. Considering that cyberbullying is widespread, is influenced by gender, and has detrimental consequences for mental health, the manuscript would benefit from a brief mention (a short paragraph would be fine) in the Introduction and then how their results may be applied to cyberbullying. It may be helpful to see, among the others:
Álvarez-García, D., Barreiro-Collazo, A., & Núñez, J. C. (2017). Cyberaggression among adolescents: Prevalence and gender differences. Comunicar, 25(50), 89–97.
Perasso, G., Carone, N., Lombardy Group 2014, H. B. I. S. A. C., & Barone, L. (2021). Written and visual cyberbullying victimization in adolescence: Shared and unique associated factors. European Journal of Developmental Psychology, 18(5), 658-677.
Wang, J., Jannotti, R. J., & Luk, J. W. (2011). Peer victimization and academic adjustment among early adolescents: Moderation by gender and mediation by perceived classmate support. Journal of School Health, 81(7), 386–392.
Author Response
Response to Reviewer 2: First, we would like to express our sincere gratitude for all comments and suggestions received from the Reviewer 2. This information has certainly enriched the text for its best understanding, thank you very much indeed. We have clarified the reviewer’s questions. We have introduced the required changes both in our answers to the specific comments and in the final manuscript. ABSTRACT: In the abstract it seems that something is missing here: "Age between 10 and 15 years old (mean=11.66, standard deviation=1.206) both males (50.5%) and females (49.5%)" Response: Thank you very much for pointing it out, we've corrected it in the Abstract. The new information is here: “A total of 562 students belonging to the 5th (n=228) and 6th (n=186) primary school year and the 1st (n=134) and 2nd (n=94) secondary school year participated in the research. They were males (50.5%) and females (49.5%) with ages between 10 and 15 years old (mean=11.66, standard deviation=1.206).” CIBERBULLYING There is no mention of cyberbullying in the Introduction, but it is only mentioned as a future direction. Considering that cyberbullying is widespread, is influenced by gender, and has detrimental consequences for mental health, the manuscript would benefit from a brief mention (a short paragraph would be fine) in the Introduction and then how their results may be applied to cyberbullying. Response: Thank you very much for pointing it out, we've included it the Introduction section. The new information is here: “It should also be noted that nowadays, bullying behaviors has spread to social networks, internet use and mobile phones; therefore, disagreements have been observed in the literature regarding the definition of bullying, linked to discipline and culture [8]. It is worth noting that cyberbullying is described as harmful acts that are intentionally carried out and inflicted through electronic devices. However, this term cannot be considered a unidimensional construct as it includes a wide range of online experiences such as exclusion, cyberbullying, denigration or impersonation [10].” “With regard to bullying at school, it is important to mention, ultimately, cyberbullying through electronic communication devices. Bullying through these media offers a very different problem to "traditional" bullying, as aggression through electronic devices helps to protect the anonymity of the aggressors and in many cases the aggressor is not aware of the consequences of their actions on the victim, making it very difficult to empathise with the victim. These aggressions can be carried out anywhere and at any time, which complicates adult supervision and control [35]. This cyber aggression has also been negatively associated, not only in victims but also in aggressors, with an increase in internalised problems such as depressive symptoms in adolescents, with serious repercussions on concentration and academic performance [36]. However, it is very difficult to determine the prevalence of cyberbullying, although published studies suggest a progressive trend [35].” REFERENCES. Response: Thank you very much for pointing it out, we have included all the references suggested. Thank you very much for all your recommendations Prof Ana Isabel Obregón Cuesta
Round 2
Reviewer 1 Report
A thorough proofread is required.
Line 157 please explain what P.A.R. is.
Line 161 - 176: Did you measure for cyber-bullying?
Line 153 (and in parts of the discussion): It is not clear where you are discussing the literature and then your study. Perhaps starting with 'Our study showed...' or 'Results from our study show.....'
Please check referencing. For example, Herrera-López (27) is 38 in the reference list.
Author Response
Response to Reviewer 1:
First, we would like to express our sincere gratitude for all comments and suggestions received from the Reviewer 1 in 2nd round. This information has certainly enriched the text for its best understanding, thank you very much indeed. We have clarified the reviewer’s questions. We have introduced the required changes both in our answers to the specific comments and in the final manuscript.
PAR program:
Response: Thank you very much for pointing it out, we've corrected it in the Material and Methods section. P.A.R is a Spanish program against bullying and intimidation. The Spanish acronyms for Stop the Bullying, Support the Victim, and Reeducate the aggressor. We have corrected it in the manuscript.
“There is a protocol of action against school bullying and a specific anti-bullying program, named SSR (Stop the bullying, Support the victim and Reeducate the aggressor, P.A.R. in Spanish) which includes actions to enhance relationships in the educational community.”
CIBERBULLYING
Response: Thank you very much for pointing it out. In the present investigation, cyberbullying has not been measured. We've included a mention of cyberbullying in the introduction and discussion. We consider that it would be very interesting to extend the study presented here in future research, including cyberbullying. The new information is here:
In terms of bullying at school, it is important to mention, ultimately, cyberbullying through electronic communication devices. Electronic bullying offers a very different problem to "traditional" bullying, as the aggression through electronic devices helps to protect the anonymity of the aggressors and in many cases the aggressor is not aware of the consequences of his actions on the victim, making it very difficult to empathize with the victim. These aggressions can take place anywhere and at any time, which complicates adult supervision and control [35]. This cyber aggression has also been negatively associated, not only in victims but also in aggressors, with an increase in internalized problems such as depressive symptoms in adolescents, with serious repercussions on concentration and academic performance [36]. However, it is very difficult to determine the prevalence of cyberbullying, although published studies suggest a progressive trend [35].
“Future research on bullying should be complemented by new forms of harassment such as cyberbullying, which is becoming increasingly prevalent.”
REFERENCES.
Response: Thank you very much for pointing it out, we have corrected all the references.
PROOFREAD
Response: Thank you very much for this comment, we have done a detailed overview of the language and references and clarified when we talk about the study and when we talk about literature.
Thank you very much for all your recommendations
Prof Ana Isabel Obregón Cuesta
